# Mechanomyographic Analysis for Muscle Activity Assessment during a Load-Lifting Task

**DOI:** 10.3390/s23187969

**Published:** 2023-09-19

**Authors:** Matthieu Correa, Maxime Projetti, Isabelle A. Siegler, Nicolas Vignais

**Affiliations:** 1Laboratoire CIAMS (Complexité, Innovation, Activités Motrices et Sportives), Université Paris-Saclay, CEDEX, 91405 Orsay, France; isabelle.siegler@universite-paris-saclay.fr (I.A.S.); nicolas.vignais@universite-paris-saclay.fr (N.V.); 2Laboratoire CIAMS (Complexité, Innovation, Activités Motrices et Sportives), Université d’Orléans, 45067 Orléans, France; 3Moten Technologies, 92800 Puteaux, France

**Keywords:** mechanomyography, electromyography, isometric contractions, dynamic contractions, load-lifting

## Abstract

The purpose of this study was to compare electromyographic (EMG) with mechanomyographic (MMG) recordings during isometric conditions, and during a simulated load-lifting task. Twenty-two males (age: 25.5 ± 5.3 years) first performed maximal voluntary contractions (MVC) and submaximal isometric contractions of upper limb muscles at 25%, 50% and 75% MVC. Participants then executed repetitions of a functional activity simulating a load-lifting task above shoulder level, at 25%, 50% and 75% of their maximum activity (based on MVC). The low-frequency part of the accelerometer signal (<5 Hz) was used to segment the six phases of the motion. EMG and MMG were both recorded during the entire experimental procedure. Root mean square (RMS) and mean power frequency (MPF) were selected as signal extraction features. During isometric contractions, EMG and MMG exhibited similar repeatability scores. They also shared similar RMS vs. force relationship, with RMS increasing to 75% MVC and plateauing to 100%. MPF decreased with increasing force to 75% MVC. In dynamic condition, $RMS_{MMG}$ exhibited higher sensitivity to changes in load than $RMS_{EMG}$. These results confirm the feasibility of MMG measurements to be used during functional activities outside the laboratory. It opens new perspectives for future applications in sports science, ergonomics and human–machine interface conception.

## 1. Introduction

Today, measuring muscle activity in ecological situations is a major issue. It could allow a better understanding of the neuromuscular system and its adaptations to the environment compared to isolated contractions in the laboratory. It may also have significant applications. In sports science, assessing muscle load may enhance performance optimization strategies, injury prevention monitoring and rehabilitation. In physical ergonomics, muscle evaluation can help reduce work-related musculoskeletal disorders through workplace assessment and reconception [1]. In human–machine interface conception, recording muscle activity in real time can be useful to control assisting device such as active exoskeletons, cobots and myoelectronic prostheses.

To effectively measure muscle activity in functional and ecological situations, the method used should be easy to set up and to use, and robust in all environments and during prolonged duration acquisitions. Today, the reference method for muscle activity measurement is electromyography (EMG), which measures the electrical activity of muscles. However, it requires skin preparation and a precise sensor placement for reproducible results [2,3], making it hard to use and to set up outside of a laboratory. Moreover, EMG is sensitive to interference and changes in impedance, making it non-robust to the environment and during extended time acquisitions, such as when sweating occurs, for example [4]. Finally, EMG is sensitive to motion artifacts because of the electrodes’ shift relative to muscle fibers in dynamic muscle actions [5,6]. Altogether, these limitations make EMG hardly suitable for recording muscle activity in functional and ecological conditions [3].

Other techniques have emerged for recording muscle activity such as mechanomyography (MMG), the mechanical counterpart of EMG, which is the measurement of the low-frequency lateral oscillations of active muscle fibers [7]. MMG reflects three physiological phenomena: (i) the gross lateral movement of the muscle at the start of the contraction, (ii) subsequent vibrations at the muscle’s resonance frequency and (iii) dimensional changes of active muscle fibers [8]. MMG can be obtained by multiple means: piezoelectric crystals, condenser microphones, displacement lasers and accelerometers [9]. Yet, accelerometers are often preferred because of their flat frequency range, ease of fixation on the skin, inexpensiveness, lightweightness and their measurement in physical units (m/s${}^{2}$). However, accelerometers are highly sensitive to motion artifacts [10]. Thus, without proper filtering or experimental design, the MMG signal could be polluted with motion artifacts, biasing its interpretation. Most studies used a band-pass filter between 5 and 100 Hz to eliminate both motion artifacts and high-frequency noise [8]. Recently, advances in MMG signal processing methods have led to more efficient motion artifact filtering in dynamic conditions [11,12]. Being sensitive to motion could also be viewed as a strong argument in favor of accelerometer-based MMG, as the low-frequency part of the signal (motion accelerations) can be used as a segmenting tool [13], making the most out of the overall signal.

MMG appears to have a number of advantages over EMG such as no skin preparation, ease of fixation and less sensitivity to sensor placement [14], making it more convenient. Moreover, it is more robust than EMG thanks to its non-sensitivity to changes in impedance and interference, thus guaranteeing signal quality over long periods of time. They can also be used in a “2 in 1” fashion, for both measuring muscle activity and segmenting postures and motions. Altogether, these technical aspects make MMG a convenient and suitable alternative to EMG for measuring and segmenting functional activities in controlled environments and ecological situations.

Despite a great number of studies regarding isometric and isokinetic contractions, very little MMG literature characterizes MMG under poly-articular functional activities such as squatting [3,13]. To the best of our knowledge, no MMG studies were found regarding load-lifting tasks, despite being omnipresent in sports, manual work and daily activities. First of all, such research could assess the validity of MMG measurements under these conditions. Moreover, it could provide significant insights about muscle synergies and neuromuscular adaptations in functional situations.

Thus, the objectives of this study are twofold: (i) comparing a new accelerometer-based MMG system to an EMG reference system during isometric contractions and (ii) analyzing a load-lifting task with both MMG and EMG to explore MMG applications during complex dynamic conditions.

## 2. Materials and Methods

### 2.1. Participants

Only male subjects were selected in this study for gender considerations [15]. Participants should not have suffered from any musculoskeletal disorders in the 6 months prior to the study. Given a significance criterion of α=0.05 and an a priori large effect size, the optimal number of participants for the current study was 18, with an actual statistical power of 0.96.

Twenty-two active males (age: 25.5 ± 5.3 years; height: 1.79 ± 0.07 m; weight: 75.7 ± 11.9 kg) participated in this study. All subjects turned out to be right-handed. Participants provided written informed consent before taking part in this experiment. This study was approved by the Academic Research Ethics Committee (Université Paris-Saclay, 2021-287).

### 2.2. Sensors

EMG signal was recorded using sensors (Miniwave, Cometa, Montegranaro, Italy) sampled at 2000 Hz and positioned following the SENIAM recommendations [2]. Prior to EMG sensor placement, the skin was shaved and sanitized with 70% alcohol-soaked wipes. MMG signal was recorded using accelerometer sensors (Moten Technologies, Puteaux, France) with a 1000 Hz sampling rate. The MMG sensor specifications are displayed in Table 1. Each sensor was placed on the muscle’s belly following recommendations from the literature [16]. A force sensor was used to measure external force at wrist level with a sampling rate of 2000 Hz (Sensor 2715-ISO, Sensy, Jumet, Belgium). Sensors were placed only on the dominant arm of each subject (see Figure 1 for EMG and MMG sensors’ location).

### 2.3. Isometric Testing Procedure

The isometric testing procedure comprised two parts: (i) the measurement of the maximum voluntary contraction (MVC) of the Biceps Brachii (BB), Triceps Brachii (TRI) and the lateral Deltoid (DEL) muscles, and (ii) the submaximal contractions at 25%, 50% and 75% of MVC for each muscle.

For the MVC measurements, participants had to perform three maximal contractions of 3 s onto the force sensor attached to an adjustable metallic structure. Trials were separated by two-minute rest periods. For the BB and the TRI muscle, the participant had the elbow flexed at 90° against the body and the force sensor was placed on the wrist. The force sensor was facing downward for the BB and upward for the TRI. For the DEL muscle, the arm was extended forward parallel to the ground with the force sensor on top of the wrist facing downward. For each muscle, the MVC value was defined as the highest value obtained across the three trials [17].

The participant then had to maintain submaximal contractions at 25, 50 and 75% of MVC for 15 s, thanks to a visual feedback of force level displayed on a screen. Each submaximal contraction was repeated three times, with a 1.5-min rest period.

### 2.4. Dynamic Testing Procedure

The dynamic testing consisted of 10 repetitions of a load lifted above shoulder level with both hands. The different weights of the load were derived from the MVC obtained for the BB muscle in the isometric condition. More precisely, the force obtained during MVC was converted into a mass to lift (25%, 50% and 75% MVC). Indeed, BB was chosen over TRI and DEL muscles as a reference because it was the strongest muscle of the three and thus allowed to have the largest load differences between intensities. Each repetition started with lifting the charge from the mid-tibia level to above shoulder level and getting back into the initial position (see Figure 2). The participant was asked to maintain a constant speed throughout every repetition. Repetition rate was set at six seconds using a metronome. Prior to the experiment, participants had a training session with an empty bar (2 kg) to practice the movement, technique and required rate.

### 2.5. Signal Processing

EMG and force signals were collected through the QTM software version 2022 (Qualisys, Göteborg, Sweden) allowing their synchronization. MMG signal was obtained using another software (Moten Launcher version 2.0, Moten Technologies, Puteaux, France). EMG and MMG signals were band-pass filtered between 20–500 Hz [5] and 5–100 Hz [16], respectively.

For the isometric testing procedure, each trial was segmented to select the 33% middle part of the contraction time according to previous studies [7,17,18,19].

The amplitudes of EMG and MMG signals were calculated by taking the root mean square (RMS) value of the segmented portion. The RMS values from EMG and MMG for each muscle were normalized from the average RMS values obtained during the three MVC trials [20]. The studied variables are referred to as $RMS_{EMG}$ and $RMS_{MMG}$. To estimate the signal frequency content, a Fast Fourier Transform was performed over the segmented data and the mean power frequency (MPF) was calculated. The studied variables are referred to as $MPF_{EMG}$ and $MPF_{MMG}$. Despite the non-stationarity of EMG and MMG signals during isometric and dynamic contractions, Fourier-based methods were shown to be acceptable during moderate velocity actions for estimating their frequency content [21].

$RMS_{MMG}$ and $MPF_{MMG}$ were computed following Equations (Equation 1) and (Equation 2), respectively [22]:(1)$$RMS_{MMG}=\sqrt{RMS_{X}^{2}+RMS_{Y}^{2}+RMS_{Z}^{2}}$$
(2)$$MPF_{MMG}=\sqrt{MPF_{X}^{2}+MPF_{Y}^{2}+MPF_{Z}^{2}}$$
where *X*, *Y* and *Z* represent the 3 axes of the accelerometer.

For the dynamic testing, the accelerometer was first low-pass filtered at 5 Hz to keep only the motion data (MOT), which is composed of low frequency accelerations [10]. We used the MOT signal of the accelerometer located on the BB muscle to compute the sensor’s pitch angle θ following Equation (Equation 3):(3)$$\theta =arctan_{2}\left(Z,\sqrt{X^{2}+Y^{2}}\right)\times -180/\pi$$
where *X*, *Y* and *Z* represent the 3 axes of the accelerometer.

θ was used to segment phases of (1) pulling the load from mid-tibia level to hip level (from posture a to b), (2) pulling the load from hip level to shoulder level (from posture b to c), (3) pushing the load above shoulder level (from posture c to d), (4) lowering the load to shoulder level (from posture d to c), (5) lowering the load to hip level (from posture c to b) and (6) lowering the load to mid-tibia level (from posture b to a), see Figure 2 and Figure 3. Like isometric testing, we used the RMS and MPF to calculate signal amplitude and frequency content over the segmented data. EMG and MMG data from dynamic contractions were not normalized since there were no dynamic MVC measurements.

### 2.6. Statistical Analysis

For the isometric testing, repeatability tests of absolute $RMS_{EMG}$ and $RMS_{MMG}$ between intensities and muscles were conducted based on intraclass correlation coefficients ICC (3, k) over the three trials for each muscle, following methodological recommendations [23]. Repeated-measures analyses of variances (ANOVA) were carried out to study the combined effects of muscles (BB/TRI/DEL) and intensities (25/50/75/100% MVC) on $RMS_{EMG}$, $RMS_{MMG}$, $MPF_{EMG}$, and $MPF_{MMG}$.

For the dynamic testing, an ICC (3, k) was conducted in order to quantify the repeatability of the MOT data over intensity levels for each muscle. Because EMG and MMG signals were not normalized in dynamic condition, two-way repeated measure ANOVAS (intensity×phase) were conducted for $RMS_{EMG}$, $RMS_{MMG}$, $MPF_{EMG}$ and $MPF_{MMG}$ separately for each muscle. We excluded phases 1 and 6 from the analysis because the recruitment of the three upper limb muscles during these phases was negligible. Post hoc analyses were conducted using the Bonferroni correction. When an intensity × phase interaction was found, we focused on the paired comparisons only between the concentric and eccentric phases of the motion where each muscle was recruited the most. For the BB muscle, phases 2 and 5 were identified as the concentric and eccentric phases, respectively (see Figure 2). For TRI and DEL muscles, phases 3 and 4 were identified as concentric and eccentric phases, respectively (see Figure 2). Average MMG and EMG RMS amplitudes in relation to % MVC were examined using a linear regression model and Pearson’s correlation test for each muscle and subject. Independent t-tests were performed to compare EMG and MMG corresponding determination coefficients $R^{2}$. All statistical analysis was conducted on JASP version 0.17. An alpha of 0.05 was considered significant for all comparisons and correlations. All data are presented as mean ± standard deviation.

## 3. Results

### 3.1. Isometric Testing

#### 3.1.1. Time Domain Analysis

The two-way repeated-measures ANOVA revealed a significant effect of intensity level on the normalized force data (F(3,63)=1855, p<0.001, η=0.98). Post hoc analysis showed that all force levels were significantly different from one another. All EMG and MMG isometric testing results are displayed in Table 2.

For both EMG and MMG, all ICC point estimates, lower and upper 95% confidence intervals were greater than 0.9. There was no significant effect of intensity level on ICC ratings. The ANOVA on $RMS_{EMG}$ exhibited a significant effect for muscle (F(2,42)=7.65, p<0.001, $\eta^{2}=0.03$), intensity (F(3,63)=270, p<0.001, $\eta^{2}=0.75$) and a significant interaction (F(6,126)=2.29, p=0.039, $\eta^{2}=0.01$). Similarly, the ANOVA on $RMS_{MMG}$ showed a significant effect for muscle (F(2,42)=6.98, p=0.001, $\eta^{2}=0.03$), intensity (F(3,63)=333, p<0.001, $\eta^{2}=0.74$) and a significant interaction (F(6,126)=7.6, p<0.001, $\eta^{2}=0.02$). Pairwise comparisons highlighted that, for all muscles, all $RMS_{EMG}$ and $RMS_{MMG}$ values were significantly different from one another, except between 75% and 100% MVC.

#### 3.1.2. Frequency Domain Analysis

Two-way ANOVA on $MPF_{EMG}$ showed a significant effect of muscle (F(2,42)=4, p=0.024, $\eta^{2}=0.11$) and intensity (F(3,63)=9.7, p<0.001, $\eta^{2}=0.06$) on $MPF_{EMG}$. Post hoc analysis revealed that $MPF_{EMG}$ at 75% MVC was lower than MPF at other intensities.

The ANOVA on $MPF_{MMG}$ yielded significant effects of muscle (F(2,42)=4.2, p<0.021, $\eta^{2}=0.07$) and intensity (F(3,63)=11.3, p<0.001, $\eta^{2}=0.11$). Post hoc analysis revealed that $MPF_{MMG}$ at 25% and 50% MVC were larger than 75% and 100% MVC.

### 3.2. Dynamic Testing

#### 3.2.1. Time Domain Analysis

The ICC results to assess the repeatability of MOT data between intensity levels for each muscle were: 0.96 for the BB, 0.99 for the DEL and 0.98 for the TRI.

All two-way repeated-measures ANOVA on $RMS_{MMG}$ and $RMS_{EMG}$ revealed a significant effect of intensity, phase and a significant intensity × phase interaction for each muscle (view Table 3).

##### Intensity Levels

Post hoc analysis highlighted that for the BB muscle, $RMS_{MMG}$ values were significantly different between intensity levels except between 25% and 50% MVC in phase 5 (eccentric contraction). For the TRI muscle, $RMS_{MMG}$ values were different between intensities except between 50% and 75% MVC in phase 3 (concentric contraction). For the DEL muscle, $RMS_{MMG}$ values were different between all intensities for both concentric and eccentric phases of contraction.

For EMG, RMS values of the BB muscle were different between 25% and 75% MVC for phase 2 and 5 (concentric and eccentric contractions); however, there were no RMS differences between 50% and 75% MVC for phase 2 and between 25% and 50% MVC for phase 5. For TRI, RMS values were only different between 25% and 75% MVC for phase 3 (concentric contraction). For DEL, RMS values were different from each other except between 50% and 75% MVC for phase 3 (concentric contraction) (see Figure 4).

##### Concentric vs. Eccentric

For MMG, there was no difference in RMS between concentric and eccentric phases for the BB muscle for all intensity levels. For the TRI and DEL muscles, RMS during the concentric contraction (phase 3) was lower than the eccentric part (phase 4) for 50% and 75% MVC.

For EMG, there was no difference in RMS between concentric and eccentric phases for the BB and TRI muscles for all intensity levels. For the DEL muscle, RMS values during its concentric contraction was greater than the eccentric contraction for all intensity levels (p<0.001).

##### EMG and MMG RMS vs. % MVC Relationship

For BB and TRI muscles, MMG $R^{2}$ were significantly greater than EMG (p=0.015 and p<0.01, respectively). For the DEL muscle, there was no difference between EMG and MMG mean determination coefficients (see Figure 5).

#### 3.2.2. Frequency Domain Analysis

All two-way repeated-measures ANOVA on $MPF_{MMG}$ and $MPF_{EMG}$ are presented in Table 4).

##### Intensity Levels

Post hoc analysis indicated that for MMG, TRI MPF decreased with increasing load. MPF at 75% MVC was significantly lower than 25 and 50% MVC. EMG MPF for the TRI and DEL highlighted similar results where MPF decreased significantly between all intensity levels.

## 4. Discussion

The aim of this study was to assess the feasibility of MMG measurements in complex dynamic situations by comparing MMG to EMG, which is the reference system to measure muscle activity, during isometric contractions and a load lifting task at different intensities. Twenty-two male participants took part in this study, consisting of submaximal isometric contractions at 25%, 50% and 75% MVC of the BB, TRI and DEL muscles. Participants then performed repetitions of a lifting task from the ground and raising it above shoulder level at three relative intensity levels. MMG and EMG signals were acquired for the Biceps, Triceps and Deltoid of the dominant arm.

### 4.1. Isometric Testing

#### 4.1.1. Time Domain Analysis

Overall, MMG repeatability scores obtained in this study can be considered excellent [24] and were higher than results previously related to MMG studies [25,26]. As an example, a relatively recent study obtained an ICC score of 0.79 with a microphone MMG [27].

$RMS_{MMG}$ results obtained in this study agree with previous investigations and the most recent MMG review article [28]. In this review, it has been shown that $RMS_{MMG}$ increases with force up to 80% MVC and reaches a plateau or even decreases to 100% MVC in isometric condition. Authors have suggested that an increase in $RMS_{MMG}$ with force is due to the increasing number of active motor units to produce force. However, at higher intensities, an increase in muscle stiffness, which is a function of attached cross-bridges and fusion of motor-unit twitches, causes an MMG amplitude plateau or a drop [28]. Similar behavior was shared with EMG, as the amplitude increases to 75% MVC and plateaus at MVC in the current study. EMG results were also in agreement with previous studies and reviews [28,29]. These results suggest that at approximately 75% MVC, the major part of motor units (MU) is recruited [28].

The evolution of MMG and EMG amplitude as a function of force level has been showed to be muscle-specific [17] but accounted only for a very small portion of the total variance ($\eta^{2}=0.02$ and $\eta^{2}=0.01$ in the current results for MMG and EMG, respectively).

#### 4.1.2. Frequency Domain Analysis

$MPF_{EMG}$ decreased from 25% to 75% MVC and increased to the initial value at 100% MVC. This result agrees with previous investigations in step isometric contraction for which authors found a similar $MPF_{EMG}$ decreasing trend from 20% to 80% MVC [30]. $MPF_{MMG}$ decreased from low intensities (25–50% MVC) to high intensities (75–100% MVC).

For both EMG and MMG, MPF correlates with the motor unit firing rate. As MPF decreased with increasing intensity, we can conclude that the increase in force was modulated by the increasing number of active motor units and by the decrease in the global motor units firing rate. Other authors suggested that muscle stiffness and/or intramuscular pressure, with increasing intensity, could impair the firing rate of motor units [31]. This impairment has been shown to stabilize or even decrease the $MPF_{MMG}$ with increasing force in step contractions.

### 4.2. Dynamic Testing

#### 4.2.1. Time Domain Analysis

Thanks to the repeatability analysis of the MOT data and the ICC scores, it was verified a posteriori that there were no significant kinematic differences between the different-load trials that could lead to misinterpretations of the MMG and EMG signals.

In the current study, we observed a significant effect of intensity, where $RMS_{EMG}$ and $RMS_{MMG}$ increased linearly with load. This result agrees with those obtained in isometric testing and in previous investigations during mono-articular isokinetic condition [7,19,32].

It was found that the $EMG_{RMS}$ of the solicited muscle was greater in the concentric part of the contraction than the eccentric part, although an inverse behavior was found for MMG RMS. Theses results agree with previous findings in isokinetic condition for which authors concluded that muscular tremor was greater during the eccentric part of the contraction [33]. These results are also consistent with those obtained in an unloaded squatting task for which the $RMS_{MMG}$ of the rectus femoris was greater entering (eccentric phase) compared to exiting the squat (concentric phase) [3].

Finally, the main result of this study concerns the fact that MMG was more sensitive to changes in load than EMG in dynamic condition. MMG amplitude exhibited greater systematic effect sizes of intensity in ANOVAs and significantly greater correlations with load for the BB and TRI muscles. A similar result has been found during incremental cycling, for which authors exhibited that “MMG amplitude more closely reflects changes in power output” [34].

#### 4.2.2. Frequency Domain Analysis

In this study, $MPF_{EMG}$ significantly decreased with increasing load for the TRI and DEL muscles. The same MPF vs. load relationship was found for MMG but only for the TRI muscle. These results agree with those obtained in isometric condition where MPF at 75% MVC was lower than 25% MVC. Moreover, MPF vs. force responses are muscle-specific, as there was no significant increase nor decrease for the BB with both EMG and MMG. This result agrees with previous studies in isokinetic condition [7,19,32].

### 4.3. Limits

One of the limits of this study is the absence of EMG and MMG normalization in dynamic conditions. Because MMG amplitude is greater in dynamic condition compared to isometric condition [35], we could not have used $RMS_{MMG}$ from MVC values as a reference to normalize signals. Another limit is that in dynamic condition, MMG may be polluted with motion artifacts despite high-pass filtering above 5 Hz [10]. Even if an ulterior verification was done on kinematics between intensity levels, artifacts such as muscular tremor or higher-frequency motion harmonics might have altered the MMG signal. In future work, to address these limits, we could improve MMG and EMG signal processing to better filter motion artifacts in dynamic conditions using EMD (Empirical Mode Decomposition)-based algorithms, which showed more effective results than band-pass filtering recently [11,12,36]. Other limits of the current study are related to gender as only male subjects have been selected. Moreover, the age of the participants (25.5 ± 5.3) may also reduce the scope of this work. To extend outcomes of this study in the future, it would be interesting to include more participants of different ages, both men and women.

## 5. Conclusions

To the best of our knowledge, this study was the first to investigate MMG and EMG during a functional load lifting task at different intensities. This study widens the scope of MMG knowledge in the context of functional activities. During isometric conditions, EMG and MMG had excellent repeatability scores, as well as similar RMS and MPF vs. force relationship. During a dynamic situation corresponding to a load-lifting task, RMS from the MMG signal was more sensitive to changes in load than EMG signal. Conversely, MPF from the EMG signal was more sensitive to changes in load. Altogether, these results sustain the feasibility of using MMG measurement in complex functional activities, such as load lifting and carrying. Moreover, EMG and MMG signals may be used together to give complementary information concerning strategies of motor unit recruitment, in both time and frequency domains. This study highlighted the fact that MMG signal may be used for practical applications such as in sports science or physical ergonomics assessment. Further work is now necessary in terms of signal processing to improve the reliability of MMG signal during ecological conditions.

## Figures and Tables

**Figure 1 sensors-23-07969-f001:**
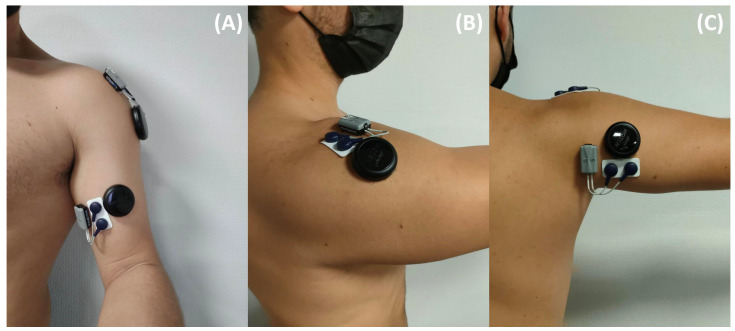
EMG and MMG sensors’ location for the long portion of the Biceps Brachii (**A**), the lateral Deltoid (**B**) and the long head of the Triceps Brachii (**C**).

**Figure 2 sensors-23-07969-f002:**
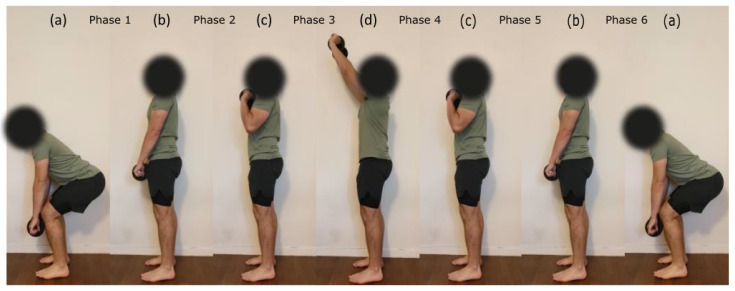
Example of the load-lifting task performed during the dynamic testing procedure. One repetition consisted of going from posture (a–d) and going back to initial posture. Postures were separated by one second from each other using a metronome. Participants were asked to keep a constant speed throughout every repetition.

**Figure 3 sensors-23-07969-f003:**
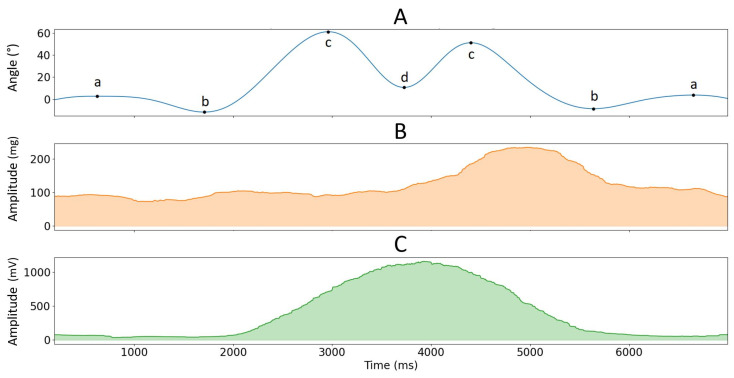
Example of the low-passed pitch angle of the accelerometer sensor located over the BB muscle (graph (**A**)) with corresponding MMG and EMG 1-second rolling RMS envelopes for the DEL muscle (graph (**B**) and graph (**C**), respectively) in function of time for one repetition of the load lifting task at 75% MVC (see Figure 2).

**Figure 4 sensors-23-07969-f004:**
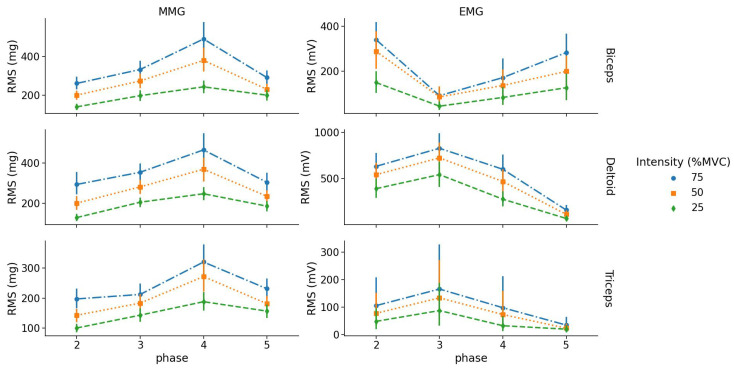
Absolute $RMS_{EMG}$ and $RMS_{MMG}$ values as a function of phase (between 2 and 5) for each intensity level and muscle (row graphics).

**Figure 5 sensors-23-07969-f005:**
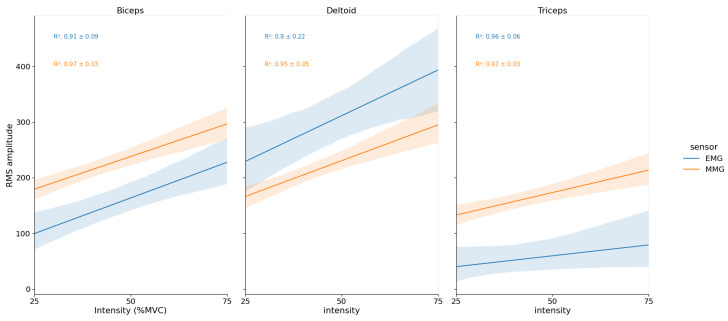
Mean EMG and MMG RMS amplitudes vs. % MVC linear relationship for each muscle during the load lifting-task. Y-axis RMS amplitude unit is in mV for EMG and in mg for MMG.

**Table 1 sensors-23-07969-t001:** MMG sensor technical specifications.

Type	3-axis capacitive accelerometer with digital output
Mass	13 g
Sensor resonant frequency	2.4 kHz
Output full scale range (FSR)	±2 g
Non linearity	0.1% FSR
Scale factor	3.9 µg/LSB (less significant bit)
Dynamic range	98 dB

**Table 2 sensors-23-07969-t002:** Isometric ICC scores and mean RMS values for each sensor, muscle and intensity level. ICC are displayed as point estimate—lower bound 95% confidence interval.

Sensor	Muscle	Feature	Intensity (%MVC)			
			25%	50%	75%	100%
EMG	BB	ICC	0.96–0.93	0.97–0.94	0.99–0.98	-
		RMS	22.1 ± 10.9 #†•	54.4 ± 23.7 *†•	80.0 ± 25.5 *#	90.2 ± 5.3 *#
	TRI	ICC	0.98–0.97	0.99–0.98	0.996–0.992	-
		RMS	15.0 ± 5.6 #†•	46.8 ± 17.6 *†•	81.7 ± 25.3 *#	89.5 ± 7.5 *#
	DEL	ICC	0.99–0.98	0.99–0.98	0.99–0.985	-
		RMS	33.8 ± 10.4 #†•	64.3 ± 18.5 *†•	92.2 ± 21.9 *#	93.6 ± 4.3 *#
MMG	BB	ICC	0.97–0.94	0.96–0.93	0.97–0.95	-
		RMS	19.4 ± 8.6 #†•	60.8 ± 25.1 *†•	81.9 ± 24.7 *#	88.8 ± 6.6 *#
	TRI	ICC	0.95–0.90	0.97–0.94	0.95–0.91	-
		RMS	11.6 ± 5.3 #†•	45.2 ± 16.7 *†•	92.5 ± 20.9 *#	85.8 ± 7.2 *#
	DEL	ICC	0.98–0.97	0.98–0.96	0.97–0.94	-
		RMS	28.3 ± 12.7 #†•	75.1 ± 20.3 *†•	100.0 ± 29.4 *#	88.3 ± 7.6 *#

*: significant difference with 25% MVC; #: significant difference with 50% MVC; †: significant difference with 75% MVC; •: significant difference with 100% MVC (*p* < 0.05).

**Table 3 sensors-23-07969-t003:** MMG and EMG RMS amplitudes two-way repeated measures ANOVA results for each muscle from dynamic testing.

Sensor	Muscle	Factor	df	F	*p*	$\eta^{2}$
MMG	BB	Intensity	(2, 36)	96	<0.001	0.40
		Phase	(3, 54)	36	<0.001	0.29
		Intensity × Phase	(6, 108)	5.5	<0.001	0.02
	TRI	Intensity	(2, 36)	84	<0.001	0.30
		Phase	(3, 54)	33	<0.001	0.35
		Intensity × Phase	(6, 108)	7.8	<0.001	0.02
	DEL	Intensity	(2, 36)	96	<0.001	0.40
		Phase	(3, 54)	36	<0.001	0.29
		Intensity × Phase	(6, 108)	5.5	<0.001	0.02
EMG	BB	Intensity	(2, 36)	37	<0.001	0.16
		Phase	(3, 54)	21	<0.001	0.32
		Intensity × Phase	(6, 108)	5.5	<0.001	0.04
	TRI	Intensity	(2, 36)	5.2	0.01	0.05
		Phase	(3, 54)	3.8	0.015	0.13
		Intensity × Phase	(6, 108)	4.58	<0.001	0.01
	DEL	Intensity	(2, 36)	32	<0.001	0.12
		Phase	(3, 54)	58	<0.001	0.59
		Intensity × Phase	(6, 108)	13	<0.001	0.02

**Table 4 sensors-23-07969-t004:** MMG and EMG MPF two-way repeated measures ANOVA results for each muscle from dynamic testing. Only significant effects are displayed.

Sensor	Muscle	Factor	df	F	*p*	$\eta^{2}$
MMG	BB	Intensity	(2, 36)	-	-	-
		Phase	(3, 54)	61	<0.001	0.67
		Intensity × Phase	(6, 108)	-	-	-
	TRI	Intensity	(2, 36)	14	<0.001	0.04
		Phase	(3, 54)	53	<0.001	0.61
		Intensity × Phase	(6, 108)	10	<0.001	0.04
	DEL	Intensity	(2, 36)	-	-	-
		Phase	(3, 54)	-	-	-
		Intensity × Phase	(6, 108)	4	<0.001	0.05
EMG	BB	Intensity	(2, 36)	-	-	-
		Phase	(3, 54)	9	<0.001	0.24
		Intensity × Phase	(6, 108)	-	-	-
	TRI	Intensity	(2, 36)	56	<0.001	0.20
		Phase	(3, 54)	17	<0.001	0.28
		Intensity × Phase	(6, 108)	7	<0.001	0.04
	DEL	Intensity	(2, 36)	14	<0.001	0.05
		Phase	(3, 54)	37	<0.001	0.55
		Intensity × Phase	(6, 108)	-	-	-

## Data Availability

The data presented in this study are available on request from the corresponding author. The data are not publicly available due to commercial privacy policy.

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
