# Peer review of "Mechanomyographic Analysis for Muscle Activity Assessment during a Load-Lifting Task"

_sensors, 2023, doi:10.3390/s23187969_

Round 1
Reviewer 1 Report
The article is well written and deals with an interesting topic.
I consider all passages – introduction, material and methods, results, discussion and conclusion – to be processed in a satisfactory manner, considering the topic addressed.
Regarding the entire article that I recommend for publication, I have only minor recommendations, mostly of a formal nature:
- L. 83-84: this sentence does not belong to the characteristics of participants, but to methods
- 22 participants were included, I am missing an indication of how the participants were selected and it would also be appropriate to add the optimal number of participants for this study, I mean the justification of the number of participants
- I recommend adding further limits to the study, especially with regard to the number of participants, age and gender
- It would be advisable to prepare the conclusion more concretely, I consider it too general, the conclusion must clearly state what the study specifically brought, it would also be appropriate to add the significance for practice
Minor editing of English required
Reviewer 2 Report
The isometric testing procedure can be described more accurately by including images (including the sensor location). Please, consider rewriting equation (3) using the "arg" function (and swapping the arguments) instead of the atan2 function, or at least rename it as "arctan2" with 2 as a sub-index.
A more thorough description of the mechanomyographic system is required to understand in more detail the signals involved.
Authors are encouraged to elaborate more on the conclusions about the ways in which both EMG and MMG techniques can become complementary.
The quality of the English language is excellent, but there is room for minor improvements. For example, authors may consider replacing the instances of "All together" with "Altogether". Similarly, a few sentences may be edited for clarity such as "above-shoulder" (include the hyphen and make it singular in every instance). I am sure a very minor revision is required, and any improvements will be a minor effort.
